# Early Regressive Development of the Subcommissural Organ of Two Human Fetuses with Non-Communicating Hydrocephalus

**DOI:** 10.3390/children9121966

**Published:** 2022-12-14

**Authors:** Emilia M. Carmona-Calero, Juan M. González-Toledo, Luis G. Hernández-Abad, Agustin Castañeyra-Perdomo, Ibrahim González-Marrero

**Affiliations:** 1Departamento de Ciencias Médicas Básicas, Facultad de Ciencias de la Salud, Campus de Ofra, Universidad de La Laguna, 38320 Santa Cruz de Tenerife, Spain; 2Instituto de Investigación y Ciencias Puerto del Rosario, 35600 Las Palmas de Gran Canaria, Spain

**Keywords:** human congenital hydrocephalus, subcommissural organ, transthyretin

## Abstract

Hydrocephalus is a central nervous system condition characterized by CSF buildup and ventricular hypertrophy. It is divided into two types: communicative and non-communicating hydrocephalus. Congenital hydrocephalus has been linked to several changes in the subcommissural organ (SCO). However, it is unclear whether these changes occur before or as a result of the hydrocephalic illness. This report presents three cases of human fetuses with hydrocephalus: one non-communicating case, two communicating cases, and two controls. Hematoxylin–Eosin (H&E) or cresyl violet and immunohistochemistry with anti-transthyretin were used to analyze SCO morphological and secretory changes. We conclude that in the cases presented here, there could be an early regression in the SCO of the communicating cases that is not present in the non-communicating case.

## 1. Introduction

The subcommissural organ (SCO) is situated between the diencephalon (epithalamus) and the mesencephalon (pretectal areas). Given that it is in contact with the third ventricle and is surrounded by a well-developed circulatory system, this structure is classified as a circumventricular organ (CVO) [1]. In humans, the SCO appears at the beginning of the eighth gestational week (GW). Its complete maturation occurs at the 15–20th GW, and the following three parts can be distinguished: the precommissural or rostral part, located in the anterior zone of the posterior commissure (PC); the subcommissural part or middle part, located under the PC; and the retrocommissural or caudal part located in the posterior zone of the PC, in the mesocoelic recess and at the beginning of the Sylvian aqueduct, and the reduction in the size of the SCO begins after the 17–20th GW, and this decrease in size begins in the precommissural, continues in the subcommissural, and finishes in the retrocommissural part [2,3,4,5,6,7,8,9,10]. The regression and atrophies of the SCO occur after birth, and the SCO disappears entirely after age 30. The SCO functions are related to the circulation and composition of the CSF, as it secretes glycoprotein into the CSF, where a fraction condenses and forms the Reissner’s fiber (RF), while the other fraction stays soluble in the CSF [7,8,11,12]. SCO cells have also been described as having a unique feature as secretory but immature neuroepithelial cells in adult mouse brains [13]. In humans, the development of the SCO is most remarkable during fetal life and produces glycoprotein, but does not form RF [5,6,7,8,11].

Hydrocephalus is a central nervous system (CNS) disorder characterized by increased CSF volume resulting in ventricular enlargement due to impaired or obstructed CSF flow [14]. Hydrocephalus is mainly classified as communicating and non-communicating [15]. In communicating hydrocephalus, the CSF flow impairment is primarily associated with subarachnoid reabsorption alterations. These alterations are due to structural blockage or reduced physiological transport at the arachnoid membrane, Pacchionian bodies, the glymphatic system, capillaries, and microvessels [16].

Ventricular zone disruption plays a fundamental role in hydrocephalus etiology, affecting the development of neural progenitor cells that protrude into the ventricular space [17]. The protrusion may obstruct this structure when this occurs at the mesencephalic aqueduct [18]. Interestingly, the SCO plays a pivotal role in maintaining the patency of the mesencephalic aqueduct [19].

Alterations in the SCO have been described in all species that develop congenital hydrocephalus. However, these works have not cleared up the critical question of whether the changes in the SCO precede hydrocephalus or are a consequence of the hydrocephalic state [6]. There is evidence that a primary defect of the SCO-Reissner´s fiber (RF) complex can lead to hydrocephalus [19].

Transthyretin (TTR) is made up of 130 amino acids that combine to form a homotetramer T4 (thyroxine) with an internal channel where T4 is attached. T4 appears to be carried over the blood–brain barrier within this complex, where it induces the manufacture of transthyretin in the choroid plexus. The protein then enters the circulation, where it attaches to T4 for delivery back to the brain [20]. Recently, new functions of the TTR related to neuronal activity at the central and peripheral nervous system levels have been described. Thus, it has been described that TTR plays an important role in brain health and nervous physiology, actions that could have an impact on certain diseases of the nervous system [21]. The TTR monomer (m) is synthesized in the choroid plexus (CP) and SCO cells and secreted into the CSF [22,23]. TTR tetramer (TTRt 45 kDa) has also been discovered in several SCO regions of the human fetus at 13 GW [23]. TTR monomer (TTRm), which is normally present in CSF but is rare or undetectable in the blood, is raised in the blood and decreased in the CSF when the blood-to-CSF barrier is disrupted [22,24].

Since the human SCO synthesizes and secretes glycoproteins into the CSF, the TTR plays an important role in cell biology function in the brain, especially in the SCO. The present work aims to examine the expression and distribution of TTR in the SCO to define if the functional activity of SCO improves or early regress associated with the type of hydrocephalus.

## 2. Methods

This study was carried out in the brains of 5 fetuses at 21 and 22 weeks of gestation, two with communicating hydrocephalus of unknown origin (H2 and H3), one obstructive hydrocephalus (Dandy–Walker) (DW H1), and two without hydrocephalus (C1 and C2). The diagnosis of hydrocephalus was established by ultrasound scan data. Ventriculomegaly (VM) was measured in the axial plane, at the level of the frontal horns and the cavum of the septum pellucidum. The calipers were positioned at the level of the inner margin of the medial and lateral walls of the atrium, at the level of the glomus of the choroid plexus, on an axis perpendicular to the long axis of the lateral ventricle.

The samples belong to the collection of the Department of Anatomy of the Faculty of Health Sciences of the University of La Laguna. Control samples C1 and C2 were spontaneous abortions at 21 and 22 weeks of gestation. The hydrocephalic fetuses were legal abortions according to Spanish legislation, H1 at 21 weeks of gestation with a VM of 14 mm, H2 at 22 weeks of gestation with a VM of 17 mm and H3 at 22 weeks of gestation with a VM of 18 mm. The mothers did not present any pathology during pregnancy.

The medical ethical committee (CEIBA2022-3205) of the University of La Laguna and the University Hospital of the Canary Islands approved the study in compliance with the legal requirements of Law 42/1988, of December 28 (Official Gazette of Spain, 12/31/1988).

Brains were extracted by a cut of the calvarium and opening of the cranial cavity; then, they were processed as previously reported [25], pieces were fixated in formaldehyde at 5%, postfixed in Bouin or Carnoy for 24 hours, dehydrated, and embedded in paraffin. Subsequently, these pieces were sectioned into sagittal sections with a thickness of 10 microns, distributed in 4 series (A, B, C, and D). Series A was stained with Hematoxylin Eosin (H-E) or cresyl violet (V) to find the SCO at the slide with the middle sagittal section and the parallel slide of the series that contain SCO (B, C, or D) was used for immunohistochemical techniques, using a rabbit polyclonal anti-prealbumin (Synonym for antigen Transthyretin, DakoCytomation, Glostrup, Denmark, code No A0002), as primary antibodies at 1:400.

Immunohistochemistry: The sections were deparaffinized, hydrated, and incubated for 18 hours in the primary antibodies anti-TTR at 1:400, diluted in Tris-Phosphate buffer, pH 7.8, containing 0.7% lambda carrageenan (Sigma-Aldrich; St. Louis, MO, USA) and 0.5% Triton X-100. The Vectastain Universal Elite ABC RTU kit (Vector Laboratories; Newark, CA, USA) was used as a secondary antibody. Negative control of the immunohistochemical technique was carried out by omitting the primary antibody. DAB (3,3-diaminobenzidine tetrahydrochloride, Sigma-Aldrich) and hydrogen peroxide were used to visualize the immunoreaction, no counteracting agents were used not to interfere with the quantification of immunoreaction. Subsequently, the samples were dehydrated in alcohol at increasing concentrations (70°, 96°, and 100°) and passed through xylene. Once these passes were finished, the tissue was mounted with Eukitt. A LEICA DMRB photomicroscope with a LEICA DC 300 F camera (Germany) was used to image the results.

## 3. Results

The subcommissural organ from the control samples showed characteristic pseudostratified epithelium with ependymal secretory cells. The ependymocytes were found as columnar cells, with euchromatin basal nuclei and a large supranuclear cytoplasm oriented toward the third ventricle. The posterior commissure and the pineal gland were fully developed at the ages studied, with its characteristic appearance distinguishing the pineal recess. The SCO was present in its pre-, sub-, and retro-commissural parts (Figure 1A,B).

Qualitatively, the SCO of the hydrocephalic samples seems unstructured, with a thinner pseudostratified epithelium where the columnar cells were not appreciated, showing reduced supranuclear cytoplasm oriented towards the ventricular cavity. The SCO is mainly present in the sub- and retro-commissural parts (Figure 1C–E).

TTR was detected in the SCO of the control cases. The TTR immunoreactivity was located at the apical pole, around the nucleus of the ependymal cells, and in the perinuclear cytoplasm of some hypendymal cells in the pre-commissural region (PRC). In the sub-commissural (SC) and retro-commissural (RC) regions, TTR immunoreactivity was located apically in scattered ependymocytes (Figure 2A,B).

In the fetus with non-communicating hydrocephalus (Figure 2C), TTR immunoreactivity was detected in the apical and perinuclear zone of the ependymocytes and the cytoplasm of some hypendymocytes of the pre, sub, and retro-commissural regions from the SCO (Figure 2C). Qualitatively, there is a remarkable TTR immunoreactivity not seen in control cases. In one of the communicating cases of hydrocephalus (Figure 2D), the TTR immunoreactivity was found in the perinuclear and apical cytoplasm of many ependymal cells at the retro-commissural area of SCO. In the other communicating case (Figure 2E), there is a noticeable TTR immunoreactivity at the apical pole of the ependymocytes in the sub- and retro-commissural areas of the SCO.

## 4. Discussion

Despite several animal studies to identify its likely causes, the genetic and/or molecular pathways that might help us comprehend the diverse forms of hydrocephalus have not yet been described [26]. The CSF route alteration in non-communicating hydrocephalus instances is localized inside the ventricular system [19]. It can be caused by a variety of diseases, including obstructions of the mesencephalic aqueduct or fourth ventricular orifices, tumors, hemorrhages, or infections within the ventricular system [26]. Because of its length and modest diameter, the midbrain aqueduct is the most occluded region [27].

Using a variety of different histological methods, it has been reported that human SCO makes its appearance at the end of the second month of gestation and its complete maturation occurs at the fifth month of gestation, when the precommissural or rostral, subcommissural or middle, and postcommissural or caudal parts of the SCO are developed [5,6,8]. The human SCO cells produce glycoproteins secreted into CSF that, unlike in animals, do not form Reissner fiber, since does not polymerize and remains soluble in the CSF [8,11]. The results presented here using H-E and cresyl violet show that the SCO is formed entirely with its pre-, sub-, and retro-commissural parts distinguished at 20 to 22 GW. This is consistent with the typical morphological development previously described in humans [4,11], which develops from the 10th to the 22nd WG.

However, several differences were found in the SCO of hydrocephalus cases showing a thinner pseudostratified epithelium and undefined columnar cells. In addition, it showed a reduced supranuclear cytoplasm oriented towards the ventricular cavity. In the pre-commissural part of the SCO of communicating cases, the reduction of the SCO was more evident, showing similar regression to what occurs several weeks later in control cases (26 GW) [4,6,8,28]. Thus, our results could indicate an early degeneration of the SCO in communicating hydrocephalus.

The human SCO at the gestational ages analyzed showed expression at TTR, and the intensity of the reaction was observed at 20 weeks of gestation in the fetus without hydrocephalus and the obstructive case, which coincides with the moments of significant development of the human SCO [6,7]. The TTR in the SCO of human embryos suggests that this ependymal gland is a source of TTR, since it is expressed in several groups of ependymal cells. TTR expression was found in the apical pole, around the nucleus, and in the perinuclear cytoplasm of several hypendymal cells in the precommissural and retrocommissural areas of the control cases and non-communicant hydrocephalus. However, in communicating hydrocephalus cases, TTR-positive cells are scarce in the pre-commissural or rostral part of the SCO. Positive cells were only found at the sub, and retrocommissural parts, which seem to be responding to an early regression of the SCO functionality in communicating hydrocephalus at 20 GW cases, since the normal regressive development of SCO corresponds to 30 GW [6,8].

From what can be seen according to the type of hydrocephalus, the variations in the SCO could be different. In this sense, the hydrocephalic HTX rat, at E18, showed little secretory activity in the middle part of the SCO, and stenosis of the mesencephalic aqueduct started, and at E19, ventricular dilatation began [29]. In this type of hydrocephalus, the malformation of the SCO precedes the hydrocephalus, since the alteration of the SCO began at E15, and the ventricular dilation started at E19 [29]. Our results did not show big differences in the SCO of the non-communicating hydrocephalus (Dandy–Walker) sample compared to controls, indicating that SCO could not play a critical role in the hydrocephalic etiology of this case. Interestingly, in communicating hydrocephalus, the alteration of the SCO is patent compared to controls, so it could contribute to the etiology of hydrocephalus. More studies are needed to confirm these results and determine whether an early involution of the SCO induces hydrocephalus, or an early regression of the SCO induces hydrocephalus.

## 5. Conclusions

There could be an early regressive development in the SCO, mainly in communicating hydrocephalus samples, that seems to not be present in the non-communicating hydrocephalus sample. This regression was confirmed at the functional level since TTR immunoreaction was also found to be scarce or absent in the frontal and middle part of the SCO in non-communicating hydrocephalus.

## Figures and Tables

**Figure 1 children-09-01966-f001:**
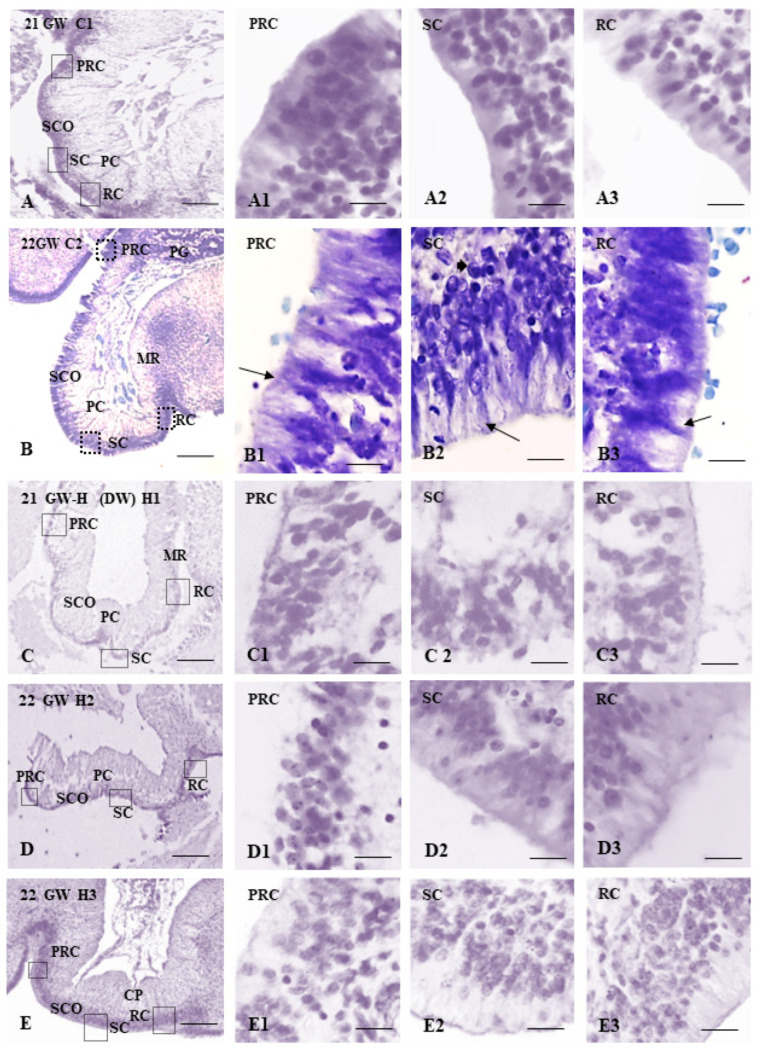
(**A**) Panoramic view of the SCO of a 21GW non-hydrocephalic control fetus (**C1**), stained with hematoxylin-eosin (bar = 250 μm). (**A1**): precommissural part of the SCO (bar = 25 μm), (**A2**): subcommissural part of the SCO (bar = 25 μm), (**A3**): retrocommissural part of the SCO (bar = 25 μm). (**B**) Panoramic view of the SCO of a 22GW non-hydrocephalic control fetus (**C2**) (bar = 250 μm), stained with cresyl violet. (**B1**): precommissural part of the SCO (bar = 25 μm), (**B2**): subcommissural part of the SCO (bar = 25 μm), (**B3**): retrocommissural part (bar = 25 μm). (**C**) Panoramic view of the SCO of a fetus with non-communicating hydrocephalus (Dandy–Walker) of 21GW (H1) (bar = 250 μm), stained with hematoxylin-eosin. (**C1**): precommissural part of the SCO (bar = 25 μm), (**C2**): subcommissural part of the SCO (bar = 25 μm), (**C3**): retrocommissural part (bar = 25 μm). (**D**) Panoramic view of the SCO of a fetus with communicating hydrocephalus of 22 GW (H2) (bar = 250 μm), stained with hematoxylin-eosin. (**D1**): precommissural part of the SCO (bar = 25 μm), (**D2**): subcommissural part of the SCO (bar = 25 μm), (**D3**): retrocommissural part (bar = 25 μm). (**E**) Panoramic view of the SCO of a fetus with communicating hydrocephalus of 22 GW (H3) (bar = 250 μm), stained with hematoxylin-eosin. (**E1**): precommissural part of the SCO (bar = 25 μm), (**E2**): subcommissural part of the SCO (bar = 25 μm), (**E3**): retrocommissural part (bar = 25 μm). For all pictures: GW = gestational week MR = mesocoelic recess, PC = posterior commissure, PG = pineal gland, PRC = precommissural or rostral region of the SCO, RC = retrocommissural or caudal region of the SCO, SCO = subcommissural organ, SC = subcommissural or middle region of the SCO, 
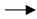
 ependymal cells, 
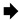
 hypendymal cells.

**Figure 2 children-09-01966-f002:**
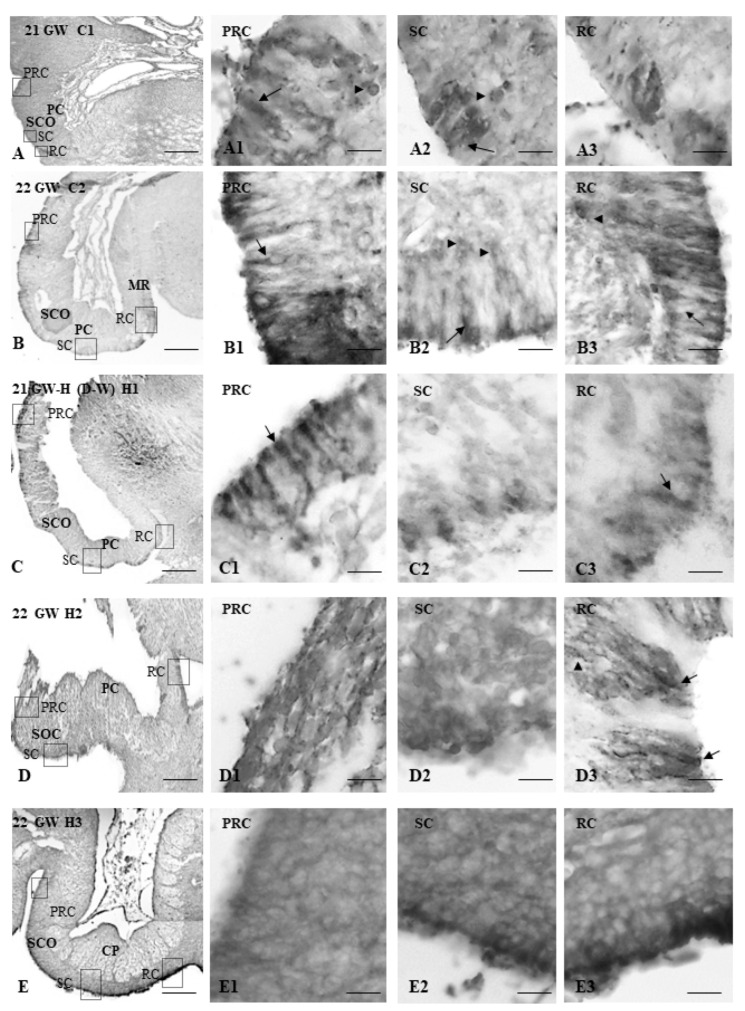
(**A**) Panoramic view of the SCO of a 21GW non-hydrocephalic control fetus (**C1**), labeled with anti-transthyretin (TTR-ir) (bar = 250 μm). (**A1**): precommissural part of the SCO (bar = 25 μm), (**A2**): subcommissural part of the SCO (bar = 25 μm), (**A3**): retrocommissural part of the SCO (bar = 25 μm). (**B**) Panoramic view of the SCO of a 22GW non-hydrocephalic control fetus (**C2**), labeled with anti-transthyretin (TTR-ir) (bar = 250 μm), (**B1**): precommissural part of the SCO (bar = 25 μm), (**B2**): subcommissural part of the SCO (bar = 25 μm), (**B3**): retrocommissural part (bar = 25 μm). In both cases controls TTR-ir is observed in group of ependymal cells and several hypendymal cells in the three parts of SCO. (**C**) Panoramic view of the SCO of a 21GW (H1) fetus with non-communicating hydrocephalus (Dandy–Walker), labeled with anti-transthyretin (TTR-ir) (bar = 250 μm), (**C1**): precommissural part of the SCO (bar = 25 μm), (**C2**): subcommissural part of the SCO (bar = 25 μm), (**C3**): retrocommissural part (bar = 25 μm). (**D**) Panoramic view of the SCO of a fetus with communicating hydrocephalus of 22 GW (H2), labeled with anti-transthyretin (TTR-ir) (bar = 250 μm), (**D1**): precommissural part of the SCO (bar = 25 μm), (**D2**): subcommissural part of the SCO (bar = 25 μm), (**D3**): retrocommissural part (bar = 25 μm). (**E**) Panoramic view of the SCO of a fetus with communicating hydrocephalus of 22 GW (H3), labeled with anti-transthyretin (TTR-ir) (bar = 250 μm). (**E1**): precommissural part of the SCO (bar = 25 μm), (**E2**): subcommissural part of the SCO (bar = 25 μm), (**E3**): retrocommissural part (bar = 25 μm). For all pictures: GW = gestational week MR = mesocoelic recess, PC = posterior commissure, PG=pineal gland, PRC = precommissural or rostral region of the SCO, RC = retrocommissural or caudal region of the SCO, SCO = subcommissural organ, SC = subcommissural or middle region of the SCO, 
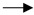
 ependymal cells, (▲) hypendymal cells. In the case of non-communicating hydrocephalus, TTR-ir can be observed in the rostral and caudal parts of the SCO. However, in the H2 case of communicating hydrocephalus, the TTR-ir can mainly be observed in the sub and retrocomisural parts of the SCO and located in the ependymal cells. In the H3 case of communicating the TTR-ir is in the ependymal cells of the retro-commissural parts, mainly in its apical pole.

## Data Availability

The datasets during and/or analyzed during the current study are available from the corresponding author on reasonable request.

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
