# Peer review of "Early Regressive Development of the Subcommissural Organ of Two Human Fetuses with Non-Communicating Hydrocephalus"

_children, 2022, doi:10.3390/children9121966_

Round 1

Reviewer 1 Report

Children 2065418 review Carmona-Calero et al.

Early regressive development of the subcommissural organ of two human fetuses with non-communicating hydrocephalus.

            The authors report the findings of 5 fetal autopsy cases, 3 with hydrocephalus and 2 controls. Their focus on the subcommissural organ (SCO) is rare and worth considering in detail. I have the following comments and suggestions:

Lines 11-22 Abstract and lines 104-105 Methods - The authors declare “one non-communicating case, two communicating cases,” but in the Methods they use the wording “two with communicating hydrocephalus with a not known origin, one obstructive hydrocephalus (Dandy-Walker),”. There is insufficient description of the autopsies to know if this is a proper separation of causes. Ideally, in tabular form, a detailed description of the brain pathology, demographics, maternal pregnancy factors, etc. must be presented. Without these, the entire manuscript fails to convincingly contrast the SCO in hydrocephalic and control fetuses.

Introduction. This section is too long. The paragraph beginning at Line 51 cut be shortened by half. 

Line 71 - the only citation pertinent to the sentence is #24 Garcia-Bonilla 2022

Line 74 - delete sentence beginning “Despite all …”

Line 83 - delete word “However”

Lines 85-91 rely on an old (1993) review. There are many more current reviews that would facilitate a coherent explanation (e.g. PMID 34297165, 32197355, 25737004)

Line 98 - incomplete sentence

Methods

Lines 104-105 - see above at Abstract. How large were the ventricles and aqueduct? I expect that the aqueduct should be enlarged in all, considering the brief description of the cases.

Line 112 “The brain pieces were processed as previously reported [34].” I looked at this publication. The methods are equally vague and the cases are non-overlapping. Much more detail is needed to convince me that the SCO was properly analyzed. The SCO is best demonstrated by a mid-sagittal slice through the thalamus / midbrain interface; is this what was done? None of the images specify the plane orientation.

Line 116 - what parts of the brains are series A-D?

Line 119 - more detail is needed about the character and source of the antibody. Is it mouse monoclonal or rabbit polyclonal? What is catalogue number from Dako (is this the A0002 rabbit PAb anti-prealbumin)?

Line 128 - was a counterstain (e.g. hematoxylin) used?

Results

Line 144 et al. Figure 1 legend, all abbreviations should be explained here (in addition to main text). Try to make the color balance even for all H&E images.

Line 154. “Worker” should be “Walker”

Figure 2 - I would like to see the color version so that the immunostaining can be better evaluated. The lettering of images is wrong (A B D E F should be  A B C D E). There are numerous arrows of different sites - they are not mentioned in the legend. A difference between the epithelium and underlying glial tissue is not always obvious. 

Line 204 “cuadal” should be caudal

Line 217 3E should be 2E

Discussion

Line 228 place a paragraph break before “The results …”

Line 242 - “The human SCO at gestational ages analyzed showed expression at TTR, …” at or of TTR? Without color images I am not convinced (see comment about Figure 2 above).

Line 259 and line 277 “From what is seen according to the type of hydrocephalus, the alteration or degeneration of the SCO is different”. I am not convinced this is true. The group sizes (2,2,1) are too small to draw any conclusions about differences.

Reviewer 2 Report

Dear Authors, 

Thank you for the opportunity to review your work. your conclusion and presentation of the histological investigation are very satisfactory. However, I didn’t find the description of the method how you obtained your samples. You suggest the references “30” but in my opinion it is not sufficient in this work. Please, clarify your method of samples winning. Your introduction is too long and has to be shorten.  

Author Response

We have corrected the method:

The study was carried out in the brains of 5 fetuses of 21 and 22 weeks of gestation, two with communicating hydrocephalus with a not known origin (H2 and H3), one obstructive hydrocephalus (Dandy-Walker) (DW H1), and two without hydrocephalus (C1 and C2). The diagnosis of  hydrocephalus was established by ultrasound scan data. The samples belong to the collection of the Department of Anatomy of the Faculty of Health Sciences of the University of La Laguna. The medical ethical committee (CEIBA2022-3205) of the University of La Laguna and the University Hospital of the Canary Islands approved the study in compliance with the legal requirements of Law 42/1988, of December 28 (Official Gazette of Spain, 12/31/1988).

Round 2

Reviewer 1 Report

Some, but not all, of my concerns have been addressed

Line 98 change to “two with communicating hydrocephalus of unknown origin”

Lines 97-101.  I am still not comfortable with the descriptions. My original recommendation now seems a necessity - A table with detailed description of the ventricle size & head circumference (from ultrasound), demographics, maternal pregnancy factors, circumstances of birth (e.g. spontaneous or induced abortion), and interval from delivery to autopsy (because autolysis can affect the immunostaining quality) must be presented. Without these, the entire manuscript fails to convincingly contrast the SCO in hydrocephalic and control fetuses.

Line 119 specify “rabbit polyclonal anti-prealbumin”

Line 128 - was a counterstain (e.g. hematoxylin) used for the immunostained slides?

Figure 1 The color balance of the H&E images has not been changed and remains poor.

Line 155. “Worker” should be “Walker” Why was this simple correction not made?

Figure 2 “We have attached color figure 2 in color, the lettering and figure legend have been corrected” - not in the manuscript version I received - it is still gray scale.

Lines 232-242I am not sure what the new  Discussion paragraph adds. The first sentence is confusing.

Lines 297-303. I am still not convinced that comparing one sample to two samples allows any conclusion, or even suggestion. We need the information about ventricle size (see above). I am more interested in hydrocephalus versus non-hydrocephalics.

Author Response

Dear reviewer, thank you for your time and suggestions. We will now respond to your comments.

1 Line 98 has been changed

2 Lines 97-101 I am still..

 Two paragraphs have been added

“Ventriculomegaly (VM) was measured in the axial plane, at the level of the frontal horns and the cavum of the septum pellucidum. The calipers were positioned at the level of the inner margin of the medial and lateral walls of the atrium, at the level of the glomus of the choroid plexus, on an axis perpendicular to the long axis of the lateral ventricle

The samples belong to the collection of the Department of Anatomy of the Faculty of Health Sciences of the University of La Laguna. Control samples C1 and C2 were spontaneous abortions at 21 and 22 weeks of gestation. The hydrocephalic fetuses were legal abortions according to Spanish legislation, H1 at 21 weeks of gestation with a VM of 14mm, H2 at 22 weeks of gestation with a VM of 17mm and H3 at 22 weeks of gestation with a VM of 18mm. The mothers did not present any pathology during pregnancy”

3 Line 119 specify “rabbit polyclonal anti-prealbumin” Has been corrected

4 Line 128

“no counteracting agents were used not to interfere with the quantification of immunoreaction”.

5 Figure 1 The color balance has been hanged, Has been corrected

6 Line 155 Walker has been corrected

7 Figure 2

The color figure 2 was attached to reviewer 1, but prefer the B&W figure 2

8 Lines 232-242 have been corrected

“Despite several animal research to identify its likely causes, the genetic and/or molecular pathways that might help us comprehend the diverse forms of hydrocephalus have not yet been described [26]. The CSF route alteration in non-communicating hydrocephalus instances is localized inside the ventricular system [19]. It can be caused by a variety of diseases, including obstructions of the mesencephalic aqueduct or fourth ventricular orifices, tumors, hemorrhages, or infections within the ventricular system [26]. Because of its length and modest diameter, the midbrain aqueduct is the most occluded region” [27].

9 Lines 293-303

“in communicating hydrocephalus, the alteration of the SCO is patent compared to controls, so it could be contributing to the etiology of hydrocephalus”

Sincerely

Reviewer 2 Report

Thank you for the realization of my suggestions. All modification are implemented.

Good work.

Author Response

Dear reviewer,  Thank you for taking the time to review my manuscript.